# 3,3′,4,5′-Tetramethoxy-trans-stilbene Improves Insulin Resistance by Activating the IRS/PI3K/Akt Pathway and Inhibiting Oxidative Stress

**Yi Tan [1], Lingchao Miao [1], Jianbo Xiao [2,*] and Wai San Cheang [1,*]**

[1] State Key Laboratory of Quality Research in Chinese Medicine, Institute of Chinese Medical Sciences, University of Macau, Avenida da Universidade, Taipa, Macau 999078, China; mc05845@um.edu.mo (Y.T.); yb97511@connect.um.edu.mo (L.M.)

[2] Department of Analytical Chemistry and Food Science, Faculty of Food Science and Technology, University of Vigo, 36310 Vigo, Spain

\* Correspondence: jianboxiao@uvigo.es (J.X.); annacheang@um.edu.mo (W.S.C.); Tel.: +853-8822-4914 (W.S.C.)

**Abstract:** The potential anti-diabetic effect of resveratrol derivative, 3,3′,4,5′-tetramethoxy-trans-stilbene (3,3′,4,5′-TMS) and its underlying mechanism in high glucose (HG) and dexamethasone (DXMS)-stimulated insulin-resistant HepG2 cells (IR-HepG2) were investigated. 3,3′,4,5′-TMS did not reduce the cell viability of IR-HepG2 cells at the concentrations of 0.5–10 μM. 3,3′,4,5′-TMS increased the potential of glucose consumption and glycogen synthesis in a concentration-dependent manner in IR-HepG2 cells. 3,3′,4,5′-TMS ameliorated insulin resistance by enhancing the phosphorylation of glycogen synthase kinase 3 beta (GSK3β), inhibiting phosphorylation of insulin receptor substrate-1 (IRS-1), and activating phosphatidylinositol 3-kinase (PI3K)/protein kinase B (Akt) pathway in IR-HepG2 cells. Furthermore, 3,3′,4,5′-TMS significantly suppressed levels of reactive oxygen species (ROS) with up-regulation of nuclear factor erythroid 2-related factor 2 (Nrf2) expression. To conclude, the beneficial effect of 3,3′,4,5′-TMS against insulin resistance to increase glucose consumption and glycogen synthesis was mediated through activation of IRS/PI3K/Akt signaling pathways in the IR-HepG2 cells, accomplished with anti-oxidative activity through up-regulation of Nrf2.

**Keywords:** 3,3′,4,5′-tetramethoxy-trans-stilbene; HepG2 cells; insulin resistance; oxidative stress; glucose consumption; glycogen synthesis

## 1. Introduction

Diabetes is a chronic metabolic disease characterized by hyperglycemia. Over time, different complications, such as vasculopathy and retinopathy, disability or even death, occur in patients through the exacerbation of diabetes. In the past 30 years, the prevalence of diabetes has kept increasing worldwide [1]. Type 2 diabetes mellitus (T2DM) accounts for about 90% of diabetes cases and is characterized by deteriorated glucose metabolism with hyperglycemia and insulin resistance (IR) [2]. Abnormal insulin signaling may result in a reduction in glucose transport and glucose phosphorylation, and it also suppresses the activity of glycogen synthase (GS) [3]. In addition, there are close associations between IR and oxidative stress [4]. A high level of reactive oxygen species (ROS) inhibits the insulin response and exacerbates the pathologic condition of diabetic patients [5]. ROS-triggered damage varies depending on the species, location of ROS and strength of feedback [6]. Defining the tipping point of ROS causing damage remains a challenging issue. At the same time, acute or chronic hyperglycemia in diabetes increases ROS production [7].

The liver plays an extremely important role in glucose metabolism, maintaining a constant blood glucose level through glycogenesis and glycogenolysis. Insulin promotes the production of liver glycogen and inhibits gluconeogenesis to reduce glucose output in the liver [8,9]. The HepG2 cell line has the functional and physiological characteristics of

normal hepatocytes, and thus, it has been used as an important model for exploring insulin resistance in vitro [10]. According to previous studies, glucocorticoids have been noted to exacerbate hyperglycemia in T2DM patients by destroying glucose homeostasis [11]. At the same time, high glucose (HG) and dexamethasone (DXMS) can induce IR, which is associated with oxidative stress [12–14], and the combined utilization of DXMS and HG has been used to induce insulin resistance in previous studies [15,16]. In this study, the in vitro insulin-resistant HepG2 (IR-HepG2) cell was established by treating high glucose and DXMS. It is well known that the insulin receptor substrate (IRS) protein family plays an important role in the process of insulin signal transduction, secretion and metabolism [14,17]. IRS1 and IRS2 are very important in insulin-mediated glucose and lipid metabolism [18], and the abundance status of IRS1 and IRS2 is related to the activation of the phosphoinositide-3-kinase (PI3K)/protein kinase B (Akt) pathway [19]. The stimulation of the PI3K/Akt signaling pathway promotes glucose uptake and activates glycogen synthase kinase 3 beta (GSK3β) to facilitate glycogen synthesis [20]. In addition, the PI3K/Akt pathway is a critical regulator in other normal cellular processes, such as growth, reproduction and apoptosis. Importantly, oxidative stress is considered a contributing factor to the onset and progression of diabetes [21]. The nuclear factor erythroid 2-related factor 2 (Nrf2), which is a key transcription factor regulating antioxidant stress, has been shown to protect cells from the damage of oxidative stress in diabetes [22]. Extensive evidence has demonstrated the anti-diabetic properties of resveratrol, trans-3,5,4-trihydroxystilbene, in animal and human studies [23,24]. 3,3′,4,5′-tetramethoxy-trans-stilbene (3,3′,4,5′-TMS) is a methoxy derivative of resveratrol. 3,3′,4,5′-TMS shows superior pharmacokinetic characteristics to remedy the shortcomings of resveratrol, including low bioavailability and a short half-life. Previous results have revealed the anti-tumor, anti-inflammatory and anti-allergic activities of 3,3′,4,5′-TMS [25–27]. However, the effect of 3,3′,4,5′-TMS on high glucose-induced IR and its molecular mechanism remains to be explored. Consequently, our present study will investigate the anti-diabetic effect and the underlying mechanism of 3,3′,4,5′-TMS in IR-HepG2 cells triggered by high glucose and DXMS.

## 2. Materials and Methods

### 2.1. Chemicals and Reagents

Dimethyl sulfoxide (DMSO) was obtained from Hangzhou Fude Biological Technology Co., Ltd. (Hangzhou, Zhejiang, China), and ethanol was purchased from Sinopharm Chemical Reagent Co., Ltd. (Shanghai, China). Dexamethasone (DXMS) and D-glucose ($\geq$99.5%) were obtained from Sigma–Aldrich (St. Louis, MO, USA). 3,3′,4,5′-TMS (purity > 98%) were acquired from Tokyo Chemical Industry Co., Ltd. (Tokyo, Japan). The primary antibodies against GAPDH, p-GSK3β (Ser9), GSK3β, p-Akt (Ser473), Akt, p-IRS1 (Ser307), IRS1, IRS2 and PI3K were obtained from Cell Signaling Technology (Danvers, MA, USA), whilst Nrf2 was supplied by Beyotime Biotechnology (Shanghai, China). The secondary anti-rabbit antibodies were obtained from Cell Signaling Technology.

### 2.2. Cell Culture

The HepG2 cells were acquired from the China Academy of Sciences (Shanghai, China). The cells were cultured with Low-glucose Dulbecco's modified Eagle's medium (DMEM) supplemented with 10% fetal bovine serum (FBS), penicillin-streptomycin and non-essential amino acids (NEAA), which were obtained from Gibco (Carlsbad, CA, USA). The cells were maintained at 37 °C in a 5% $CO_2$ humidified incubator. In order to establish the IR cell model, the HepG2 cells were pretreated with DXMS (30 µM, dissolved in ethanol) plus D-glucose (50 mM final concentration) for 48 h. Thereafter, the experimental groups were treated with different concentrations (1 µM and 2.5 µM) of 3,3′,4,5′-TMS for 16 h. The group not treated with DXMS, D-glucose or 3,3′,4,5′-TMS served as the control.

### 2.3. MTT Assay

The cytotoxic effects of 3,3′,4,5′-TMS and DXMS on the HepG2 cells were measured by the 3-(4,5-dimethylthiazol-2-yl)-2,5-diphenyltetrazolium bromide (MTT) assay. MTT reagent was obtained from Sigma-Aldrich (St. Louis, MO, USA). According to the protocol provided by the MTT supplier, HepG2 cells ($7 \times 10^3$ cells/well) were seeded on 96-well culture plates and cultured overnight. Then, the cells were incubated with different concentrations of 3,3′,4,5′-TMS and DXMS for 48 h. At the same time, ethanol and DMSO were used as vehicle to treat the cells for 48 h at the maximum administration volume of 3,3′,4,5′-TMS and DXMS. A group with no treatment served as control. The time of incubation for 48 h in the MTT assay was selected as HepG2 cells were exposed to DXMS plus D-glucose to induce insulin resistance in latter experiments. The number of cells seeded was selected after screening the different cell densities according to the linearity range at which no cell death was observed. After incubation, the cells were refreshed with 10% MTT-containing medium at 37 °C for 3 h. At last, the supernatants were discarded, and 150 μL of DMSO was added to each well to dissolve the formazan crystals by shaking the culture plate for 30 min. The absorbance was read at 570 nm using a SpectraMax M5 microplate reader (Molecular Devices, Silicon Valley, CA, USA). The relative cell viability was calculated as the percentage of control group.

### 2.4. Glucose Consumption Assay

The effects of 3,3′,4,5′-TMS on glucose consumption potency were determined by a glucose assay kit, which was obtained from Nanjing Jiancheng Bioengineering Institute (Nanjing, China, A154-1-1). HepG2 cells ($7 \times 10^5$ cells/well) were seeded on 6-well culture plates and cultured overnight. The cells were treated with DXMS plus D-glucose for 48 h and different concentrations of 3,3′,4,5′-TMS for 16 h. According to the manufacturer's instructions, the supernatants were collected, and their glucose concentrations were measured by the kit. The contents of cell glucose were determined by the glucose oxidase method, which causes an amaranth tint in the sample. The absorbance at 505 nm was detected with a microplate spectrophotometer. The result was calculated as below: glucose consumption = glucose concentration of blank–glucose concentration of each treatment group.

### 2.5. Glycogen Synthesis Assay

The effect of 3,3′,4,5′-TMS on glycogen synthesis ability was determined by a glycogen assay kit (Nanjing Jiancheng Bioengineering Institute, Nanjing, Jiangsu, China, A043-1-1). HepG2 cells ($7 \times 10^5$ cells/well) were seeded on 6-well culture plates and cultured overnight. Then, the cells were treated with DXMS, D-glucose and different concentrations of 3,3′,4,5′-TMS as previously described in Section 2.2. According to the manufacturer's instructions, the cells were collected in phosphate-buffered saline solutions with cell disruption by ultrasonication. After further hydrolysis in alkali solution, the contents of cell glycogen were determined by sulfuric acid–anthrone method, which causes a blue tint in the sample. The absorbance at 620 nm was detected with a microplate spectrophotometer. The cell glycogen levels were calculated by comparing with the control group.

### 2.6. Western Blot Assay

HepG2 cells ($7 \times 10^5$ cells/well) were seeded on 6-well culture plates and cultured overnight. Treatments of the cells were the same as above in Section 2.2. After treatment, the medium was discarded, and the residual medium was washed away by phosphate-buffered saline (PBS). The cells were harvested and lysed with RIPA solution containing 1% Protease Inhibitor Cocktail and 1% phenylmethanesulfonyl fluoride (PMSF), both of which were obtained from Beyotime Biotechnology. The above operations were all performed on ice. Subsequently, the cell lysates were centrifuged at 15,000 rpm for 30 min at 4 °C to collect supernatants. The BCA Protein assay kit (Beyotime Biotechnology, Shanghai, China) was used to measure the total protein contents. Protein samples were separated

by 8–10% SDS/PAGE gels, and all materials for SDS–PAGE were obtained from Bio-Rad (Hercules, CA, USA). Afterwards, the proteins were transferred to PVDF membranes (Millipore, Billerica, MA, USA). The membranes were blocked by 5% non-fat milk (Bio-Rad) for 2 h, followed by incubation with the appropriate primary antibodies overnight at 4 °C. After washing with Tris Buffered Saline with Tween-20 (TBST) buffer, the membranes were incubated with the secondary antibodies for 2 h at room temperature. Finally, the protein bands were detected by enhanced chemiluminescence (ECL) reagent purchased from Thermo Fisher Scientific (Waltham, MA, USA) and ChemiDoc MP Imaging System (Bio-Rad Laboratories, Hercules, CA, USA).

### 2.7. ROS Assay

The intracellular ROS of HepG2 cells was determined by 5-(and-6)-chloromethyl-2′,7′-dichlorodihydrofluorescein diacetate acetyl ester (CM-H$_2$DCFDA) [28] that was acquired from Invitrogen (Carlsbad, CA, USA) and Leica-DMi8 Inverted fluorescent microscope (Leica Miscrosystems, Wetzlar, Germany). HepG2 cells ($5 \times 10^4$ cells/well) were seeded on 24-well culture plates and cultured overnight. After treatment with DXMS, D-glucose and different concentrations of 3,3′,4,5′-TMS, the cells were washed by PBS. Incubating with normal physiological saline solution (NPSS) that contained 10 μM CM-H$_2$DCFDA at 37 °C for 30 min in the dark, the fluorescence images were captured by Leica-DMi8 at 488/525 nm (excitation/emission). Image J software (National Institutes of Health, Bethesda, MD, USA) was used to quantify the fluorescence intensity.

### 2.8. Statistical Analysis

In this study, all the values were shown as mean $\pm$ S.E.M from at least three independent experiments. One-way analysis of variance (ANOVA) was used to analyze the differences among groups by GraphPad Prism (GraphPad Software Inc., La Jolla, CA, USA). The Newman-Keuls multiple comparison tests were applied for post hoc pairwise comparisons. A $p < 0.05$ indicated statistically significant difference.

## 3. Results

### 3.1. The Effect of 3,3′,4,5′-TMS and DXMS on the HepG2 Cell Viability

MTT assay was used to determine the effects of 3,3′,4,5′-TMS and DXMS on cell viability. The results showed that 3,3′,4,5′-TMS did not show cytotoxic effect at low concentrations (0.5–10 μM) but significantly decreased the cell viability of HepG2 cells at high concentrations (25 and 50 μM) (Figure 1a). Moreover, DXMS, as well as the solvents (0.25% *v/v* DMSO or 0.3% *v/v* ethanol), had no negative effect on the cell viability of HepG2 cells (Figure 1b).

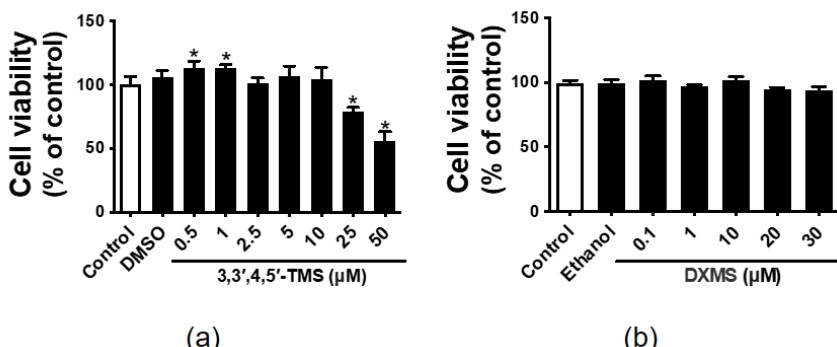

(a)  (b)

**Figure 1.** The effects of 3,3′,4,5′-tetramethoxy-trans-stilbene (3,3′,4,5′-TMS) and dexamethasone (DXMS) on cell viability in the HepG2 cells. The cell viability after treated with different concentrations of (**a**) 3,3′,4,5′-TMS and (**b**) DXMS in the HepG2 cells for 48 h. Values are the means $\pm$ SEM (*n* = 3); * $p < 0.05$ vs. Control.

### 3.2. Effect of 3,3′,4,5′-TMS on Glucose Consumption

To investigate the potential effect of 3,3′,4,5′-TMS on glucose metabolism in IR-HepG2 cells, the glucose concentration of cell culture supernatants was detected by the glucose assay kit. Compared with the control group (mean = 1.735 mM), glucose consumption of the high glucose-DXMS-treated group was significantly reduced with a mean value of 0.745 mM. Conversely, the IR-HepG2 cells significantly consumed more glucose to a level comparable with control after the treatment of 3,3′,4,5′-TMS at 2.5 μM (Figure 2a). The mean values of 1 and 2.5 μM 3,3′,4,5′-TMS were 1.018 and 1.564 mM, respectively.

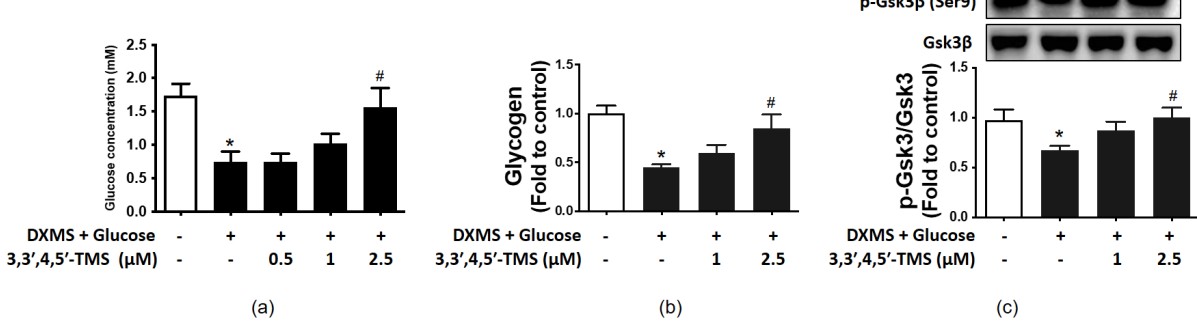

**Figure 2.** The effects of 3,3′,4,5′-TMS on glucose consumption of insulin-resistant HepG2 cells (IR-HepG2 cells). (**a**) Glucose consumption, (**b**) glycogen content and (**c**) phosphorylation of Gsk3β at Ser9 in IR-HepG2 cells pre-treated with high glucose (50 μM) plus DXMS (30 μM) for 48 h and treated with 3,3′,4,5′-TMS for another 16 h. Values are the means ± SEM ($n = 3$); * $p < 0.05$ vs. Control; # $p < 0.05$ vs. DXMS + glucose.

### 3.3. 3,3′,4,5′-TMS Enhanced Glycogen Synthesis Ability by Up-Regulating p-GSK3β

As shown in Figure 2b, treatment of high glucose and DXMS diminished the intracellular glycogen content in IR-HepG2 cells with a mean value of 0.448, whereas 3,3′,4,5′-TMS gradually elevated glycogen content in a concentration-dependent manner. The mean values of 1 and 2.5 μM 3,3′,4,5′-TMS were 0.599 and 0.850, respectively. To further explore the role of 3,3′,4,5′-TMS in glycogen synthesis, the protein expression of total GSK3β and serine phosphorylation of GSK3β (p-GSK3β at Ser9) was determined by Western blot. Compared with the control group, p-GSK3β expression in the high glucose-DXMS-induced IR-HepG2 cells was significantly down-regulated with a mean value of 0.678; and such inhibition was reversed by 3,3′,4,5′-TMS (2.5 μM, 16 h) (Figure 2c). The mean values of 1 and 2.5 μM 3,3′,4,5′-TMS were 0.873 and 1.002, respectively.

### 3.4. 3,3′,4,5′-TMS Activated IRS/PI3K/Akt Pathway

To further explore the action mechanism of 3,3′,4,5′-TMS modulating IR, Western blotting was performed to measure protein expressions of IRS/PI3K/Akt pathway, including phosphorylation of IRS1 at Ser307 and Akt at Ser473 as well as the expression of IRS1, IRS2, PI3K and Akt. IRS1 and IRS2 are the foundation of insulin signaling. High glucose and DXMS treatment attenuated the protein expressions of IRS1 (mean = 0.666) and IRS2 (mean = 0.745) and increased the phosphorylation of IRS1at Ser307 (mean = 1.925); such inhibition of insulin signaling was improved by 3,3′,4,5′-TMS in the IR-HepG2 cells (Figure 3a–d). Likewise, the PI3K/Akt signaling pathway was suppressed by the treatment of high glucose and DXMS: downregulated PI3K (mean = 0.457) expression and reduced phosphorylation of Akt at Ser473 (mean = 0.679). These changes were remarkably prevented by 3,3′,4,5′-TMS (Figure 3e,f).

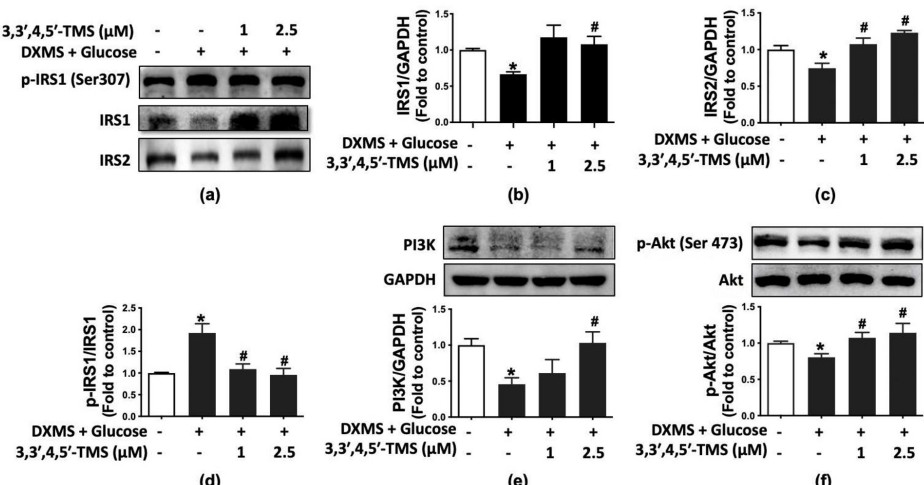

**Figure 3.** The effects of 3,3′,4,5′-TMS on IRS/PI3K/Akt signaling pathways. (**a**) Representative Western blots and summarized data for protein expressions of (**b**) IRS1, (**c**) IRS2 and (**d**) phosphorylated IRS1 at Ser307 in HepG2 cells with different treatments. (**e**) PI3K expressions and (**f**) phosphorylation of Akt at Ser473 in IR-HepG2 cells. Values are the means $\pm$ SEM ($n = 4$); * $p < 0.05$ vs. Control; # $p < 0.05$ vs. DXMS + glucose.

### 3.5. 3,3′,4,5′-TMS Alleviated Oxidative Stress by Upregulating Nrf2

There is a close linkage between oxidative stress and IR. The ROS generation of different groups was evaluated by CM-H$_2$DCFDA to investigate the antioxidant capacity of 3,3′,4,5′-TMS. In IR-HepG2 cells, the fluorescence intensity was stronger than in the control group, and the elevated ROS level was significantly reduced by 3,3′,4,5′-TMS at both concentrations (1 and 2.5 μM) (Figure 4a,b). Additionally, 3,3′,4,5′-TMS increased the Nrf2 level in high glucose-DXMS-induced IR-HepG2 cells (Figure 4c). All these results supported the antioxidant capacity of 3,3′,4,5′-TMS.

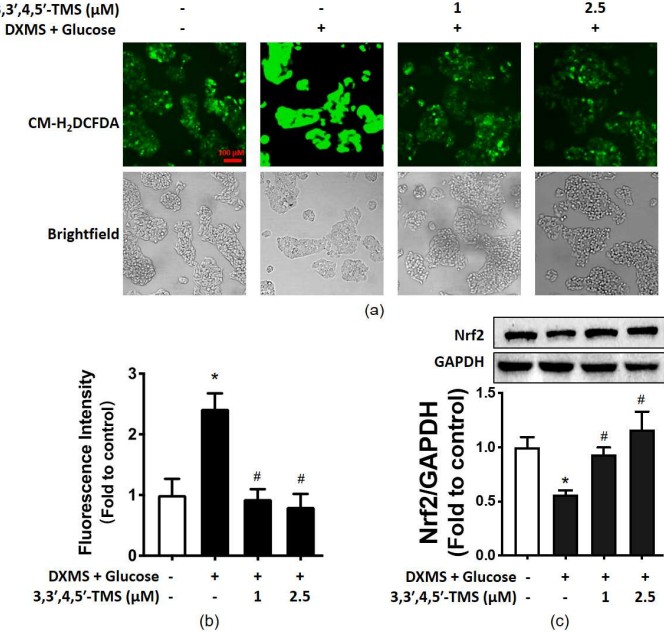

**Figure 4.** The effects of 3,3′,4,5′-TMS on oxidative stress. (**a**) Representative images and (**b**) summarized data showing CM-H$_2$DCFDA intensity in IR-HepG2 cells pre-treated with high glucose (50 mM) and DXMS (30 μM) for 48 h and treated with 3,3′,4,5′-TMS for another 16 h. (**c**) Protein expressions of Nrf2 in IR-HepG2 cells. Values are the means $\pm$ SEM ($n = 3$); * $p < 0.05$ vs. Control; # $p < 0.05$ vs. DXMS + glucose.

## 4. Discussion

In the present work, using high glucose-DXMS-stimulated IR-HepG2 cells, the methoxyl derivative of resveratrol 3,3′,4,5′-TMS (2.5 μM) demonstrated anti-diabetic potential as supported by the findings: (1) improving the hepatic glucose consumption; (2) enhancing glycogen synthesis by upregulating phosphorylated GSK3β at Ser9; (3) inhibiting phosphorylation of IRS1 at Ser307 with normalized IRS1and IRS2 levels to activate PI3K/Akt pathway; and (4) decreasing the levels of ROS by upregulation of Nrf2.

With the advances in knowledge on the pathophysiology of diabetes, IR has captured more and more attention in T2DM. Normal signal transduction of insulin promotes glucose transportation from blood to different cells, stabilizing blood glucose concentration at a normal level. In the liver, insulin activates glycogen synthesis and glycolysis and inhibits gluconeogenesis [29]. Glucose and DXMS were used together to induce IR in HepG2 cells. DXMS not only can induce IR but also can exacerbate IR induced by high glucose [15]. Combination treatment with high glucose and DXMS has been proven to be an effective method for establishing insulin resistance models [16]. DXMS causes damage to mitochondrial function, leading to exaggerated levels of ROS in mitochondria [30]. Insulin resistance involves complex mechanisms and processes, including oxidative stress [31,32]. The current study showed that 3,3′,4,5′-TMS at high concentrations (25 and 50 μM) significantly lowered the cell viability of HepG2 cells. Low concentrations (0.5–10 μM) of 3,3′,4,5′-TMS had no effect on cell viability. This result was consistent with the previous findings, which revealed the anti-tumor potency of this resveratrol derivative [33].

In T2DM, IR results in a reduction in glucose consumption in hepatocytes [34], modulating glucose transport activity and glucose metabolism [35]. Specific transporter proteins are required for facilitated diffusion of glucose into cells, such as glucose transporters (GLUTs) and sodium-glucose cotransporters (SGLTs) [36]. GLUT2 has been shown to be down-regulated in IR models to reduce glucose consumption in the IR-HepG2 cells [37–39]. After insulin stimulates the transfer of glucose to cells in the liver and muscle, glucose is converted into pyruvate through glycolysis, which supplies energy to mitochondria through the tricarboxylic acid cycle (TCA) [40]. However, hepatic glycolytic activity and glucose oxidation are impaired in T2DM [41]. In this study, we found that 3,3′,4,5′-TMS (2.5 μM) was effective to increase glucose consumption and glycogen synthesis with enhanced serine phosphorylation of GSK3β in IR-HepG2 cells. Lower concentration at 1 μM showed minor but non-significant improvement. Several studies have reported that insulin induces the inactivation of GSK3β. GSK3β becomes inactivated by phosphorylation at Ser9; this inactivation subsequently reduces the inhibitory phosphorylation of glycogen synthase (GS), activating GS and thereby promoting glycogen synthesis [42].

The PI3K/Akt pathway is involved in the mechanism of insulin transduction [43]. In general, insulin is secreted by the pancreas and then binds with the insulin receptor on the cell surface, triggering an intracellular cascade. After insulin binding to a receptor, IRS proteins are recruited to stimulate the PI3K/Akt pathway downstream [44], regulating glucose uptake, glucose synthesis, gluconeogenesis, protein synthesis, cell growth and differentiation [43]. IR is associated with decreased IRS1 and IRS2 activities, which, in turn, inactivate PI3K/Akt signaling [45]. Notably, deficiency or knockout of IRS2 can lead to IR and activate gluconeogenesis in HepG2 cells [46]. Our results showed that 3,3′,4,5′-TMS modulated the IRS/PI3K/Akt pathway to ameliorate IR and T2DM potentially.

Oxidative stress is related to the pathology of T2DM, inducing glucose intolerance, IR and β-cell dysfunction, etc. [47]. ROS activates signal pathways such as c-Jun NH2 terminal kinase (JNK), nuclear factor-κB (NF-κB), and p38 mitogen-activated protein kinase (MAPK). Of note, the activated JNK signaling pathway inhibits IRS1 and thereby inhibits downstream pathways [32]. The impact of JNK-IRS1/PI3K pathways on glucose metabolism was studied in DXMS-stimulated IR-HepG2 cells [48]. On the other hand, Nrf2, as a key transcription factor regulating oxidative stress, has become a new anti-diabetic target [49]. The expression of Nrf2 was significantly down-regulated in T2DM mice and high glucose-induced IR-HepG2 cells [50,51]. Additionally, importantly, the increase in

Nrf2 expression improves insulin sensitivity and glucose homeostasis [52]. In the liver of high-fat-diet mice, Nrf2 can be activated through the PI3K/Akt signaling pathway [53]. Nrf2 also has a potential connection with GSK3β and PI3K/Akt pathway [54,55]. Resveratrol is an Nrf2 agonist to upregulate the expression of Nrf2 [52,56]. In consistency, our study showed that 3,3′,4,5′-TMS, a resveratrol derivative at 1 µM and 25 µM reduced intracellular ROS levels and enhanced Nrf2 expression in IR-HepG2 cells.

## 5. Conclusions

In conclusion, this study is the first to show that 3,3′,4,5′-TMS could improve IR in HepG2 cells by inhibiting oxidative stress and activating the IRS/PI3K/Akt pathway, indicating the potential of 3,3′,4,5′-TMS for the treatment of diabetes (Figure 5). The effect of 3,3′,4,5′-TMS in diabetes still needs to be further studied in vivo.

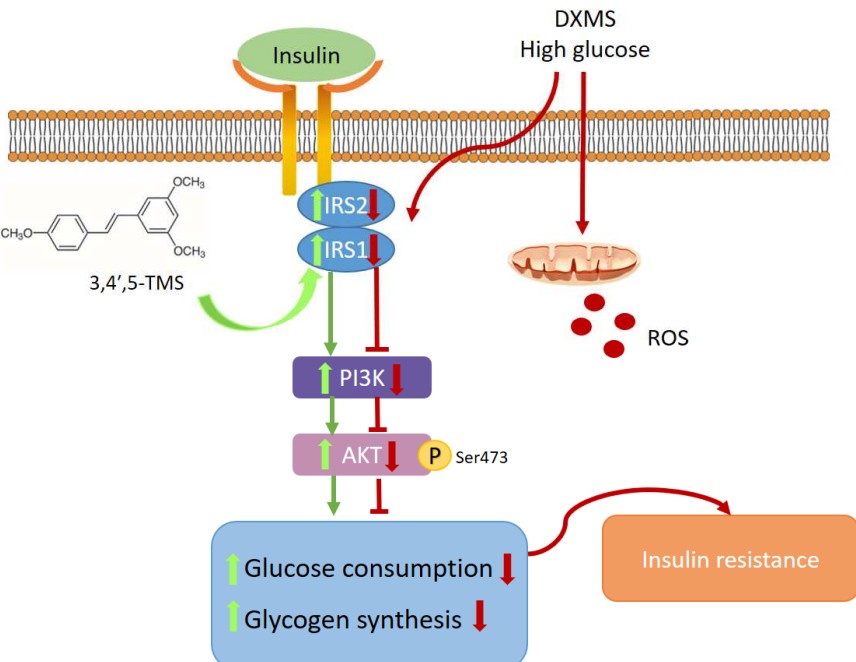

**Figure 5.** Signaling pathway conclusions. 3,3′,4,5′-TMS ameliorated insulin resistance by suppressing reactive oxygen species (ROS) production and activating insulin receptor substrate (IRS)/phosphatidylinositol 3-kinase (PI3K)/protein kinase B (Akt) pathway in IR-HepG2 cells.

**Author Contributions:** Conceptualization, W.S.C. and J.X.; methodology, Y.T. and L.M.; formal analysis, Y.T.; writing—original draft preparation, Y.T.; writing—review and editing, W.S.C. and J.X.; supervision, W.S.C.; funding acquisition, W.S.C. All authors have read and agreed to the published version of the manuscript.

**Funding:** This research was funded by Ramón y Cajal grant, grant number RYC2020-030365-I; University of Macau, grant number SRG2019-00154-ICMS, MYRG2019-00157-ICMS and MYRG2018-00169-ICMS; and the Science and Technology Development Fund of Macau (FDCT), grant number 0098/2020/A, 0117/2020/A and SKL-QRCM(UM)-2020–2022.

**Institutional Review Board Statement:** Not applicable.

**Informed Consent Statement:** Not applicable.

**Data Availability Statement:** The data presented in this study are available on request from the corresponding author.

**Acknowledgments:** The authors thank the technical team of the Institute of Chinese Medical Sciences at the University of Macau for their valuable assistance.

**Conflicts of Interest:** The authors declare no conflict of interest.

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
