# Peer review of "3,3′,4,5′-Tetramethoxy-trans-stilbene Improves Insulin Resistance by Activating the IRS/PI3K/Akt Pathway and Inhibiting Oxidative Stress"

_cimb, doi:10.3390/cimb44050147_

Round 1
Reviewer 1 Report
Dear Authors,
The main criticisms of your work are:
-The lack of correlation between all data: multivariate analysis should be used the scarce discussion of the presented results.
In addition, I list below other suggestion to improve the quality of your work.
Introduction
- This part of the work should justify the purpose of the research. It should also indicate the novelty of the research.
Develop the sentence L31-L32
- L39, what is the exact level of ROS able causing the insulin inhibition response and exacerbates the pathologic condition of diabetic patients?
- Delete the fig 1. When this structure was not elucidated by authors
Material and methods section
- Please use the multivariate analysis
Results and Discussion section?
-the results required more depth
The conclusion should be improved.
- A grammar check could be useful.
Author Response
The main criticisms of your work are:
-The lack of correlation between all data: multivariate analysis should be used the scarce discussion of the presented results.
Response: We thank the reviewer for this comment. We have added Newman-Keuls multiple comparison tests applied for post hoc pairwise comparisons in section 2.8 Statistical Analysis. Our main objective was to explore the protective effects of 3,3',4,5'-TMS against insulin resistance. Based on literature review, one-way ANOVA was used to analyze the differences among groups and post hoc tests were performed to support that 3,3',4,5'-TMS can upregulate p-GSK3β, IRS/PI3K/Akt pathway and Nrf2, improving hepatic glucose consumption, enhancing glycogen synthesis and decreasing oxidative stress. Multivariate analysis was not performed as we were not assessing multiple dependent variables simultaneously. We were not determining relationships and patterns among large sets of data.
In addition, I list below other suggestion to improve the quality of your work.
Introduction
- This part of the work should justify the purpose of the research. It should also indicate the novelty of the research.
Response: We thank the reviewer for this comment. As suggested by the reviewer, we added more information to justify the purpose of the research in Introduction. The novelty of present study is to disclose the anti-diabetic effect of 3,3’,4,5’-TMS in insulin resistant HepG2 cells.
“Extensive evidence has demonstrated the anti-diabetic properties of resveratrol, trans-3,5,4-trihydroxystilbene, from animal to human studies [22,23]. 3,3’,4,5’-tetramethoxy-trans-stilbene (3,3’,4,5’-TMS) is a methoxy derivative of resveratrol. 3,3’,4,5’-TMS was suggested with superior pharmacokinetic character-istics to remedy the shortcomings of resveratrol including low bioavailability and short half-life. Previous results have revealed the anti-tumor, anti-inflammatory and an-ti-allergic activities of 3,3’,4,5’-TMS [24-26]. However, the effect of 3,3’,4,5’-TMS on high glucose-induced IR and its molecular mechanism remains to be explored. Consequently, our present study will investigate the anti-diabetic effect and the underlying mechanism of 3,3’,4,5’-TMS in IR-HepG2 cells triggered by high glucose and DXMS.”
Develop the sentence L31-L32
Response: We thank the reviewer for this comment. We have revised as “Diabetes is a chronic metabolic disease characterized by hyperglycemia. Over time, different complications such as vasculopathy and retinopathy, disability or even deaths are resulted in patients by exacerbation of diabetes.”
- L39, what is the exact level of ROS able causing the insulin inhibition response and exacerbates the pathologic condition of diabetic patients?
Response: We thank the reviewer for this comment. Body injury caused by ROS has been extensively studied, but defining the tipping point of ROS causing damage has always been an important challenge in the field. Damage produced by ROS is affected both by the species and the location of the ROS and depends on strength of feedbacks. Therefore, we recognize this limitation should be mentioned in the paper, so we added the following sentence “ROS-triggered damage varies, depending on the species, location of ROS and strength of feedbacks [6]. Defining the tipping point of ROS causing damage remains a chal-lenging issue” in the revised manuscript.
- Delete the fig 1. When this structure was not elucidated by authors
Response: We thank the reviewer for this comment. We deleted Figure 1 as suggested by the reviewer.
Material and methods section
- Please use the multivariate analysis
Response: We thank the reviewer for this comment. We have added Newman-Keuls multiple comparison tests applied for post hoc pairwise comparisons.
Results and Discussion section?
-the results required more depth
Response: We thank the reviewer for this comment. We have added the mean values in the text.
The conclusion should be improved.
Response: We thank the reviewer for this comment. We have revised as “In conclusion, this study is the first to show that 3,3',4,5'-TMS could improve IR in HepG2 cells by inhibiting oxidative stress and activating IRS/PI3K/Akt pathway, indi-cating the potential of 3,3',4,5'-TMS for the treatment of diabetes (Figure 5). The effect of 3,3',4,5'-TMS in diabetes still needs to be further studied in vivo.”
- A grammar check could be useful.
Response: We thank the reviewer for this comment. We checked and corrected the grammatical mistakes in the revised manuscript.
Reviewer 2 Report
This is good work done by the authors. Please consider the following suggestions:
Line 33: Kindly consider including the relevant reference here.
Line 45: Kindly consider rephrasing “the drug” as no particular drug has been reported in the previous sentences. Also here the authors may want to further elaborate on why the choice of hepatocellular carcinoma cells is also appropriate for the evaluation of IR based on previous research.
Line 46-47: Please rephrase this sentence to make sure it is grammatically correct and conveys the authors’ intended message.
Line 64-66: Please rephrase this sentence to make sure it is grammatically correct and conveys the authors’ intended message.
Line 96: Kindly consider including the concentrations evaluated in this study. It would be beneficial for the reader to have this information right at the beginning in order to properly follow the development of the manuscript.
Line 100-111: Kindly include the original reference of the protocol employed by the authors (in case this is a protocol provided by the MTT supplier please consider including this information as well). It is essential for the reader to be able to understand that this is not a newly developed protocol for the evaluation of cell viability.
Line 104: Kindly elaborate on the selection on the 48h treatment as well as the number of cells seeded in each well (Considering that the volume of each well is approx.. 300μl and there is no report that the cell medium is refreshed within this 48h treatment, did the authors observe cell death ought to the number of cells increasing in each well?). Maybe a brief elaboration of the choices made on this protocol would be beneficial for the reader as well as future research.
Line 111: Kindly consider including the calculations made for the evaluation of cell viability.
Line 115 and 126: Please consider including the product number of the kit used.
Line 119 and 130: kindly consider describing the process in brief for the reader to understand what was done.
Line 130: Also include any calculations made for the final evaluation as well as the basis of the calculations (are they made against the control?)
Line 135: Also here please include the specific treatments employed.
Line 133: The authors may want to consider including relevant references to the protocols employed if they are not provided by the supplier of kits used. In case this method is reported for the first time and no previous reference is available, please include all steps that led to the method’s development and standardization.
Line 153: Please include the positive control used for this evaluation. Also, the comment regarding line 133 applies here, so the authors are kindly invited to provide the reference of the original protocol or the steps that led to the method’s development and standardization if this is a newly introduced method.
Line 176: Since both ethanol and DMSO are known to be toxic in cells in certain concentrations. Please include the volume of DMSO and ethanol used in this assay.
Line 182, 195 and 204: Please include the mean (SE) values in the text as the graphs are not very helpful for the reader to obtain this information.
Line 216: Figure 4 (a) In the image for IRS1 the lines seem to be bending. Kindly revisit.
Results: Kindly elaborate on the limitation of the concentrations evaluated after the MTT assay. As the authors stated 0.5-10μM were not toxic, but further on only 1 and 2.5 are evaluated. Please clarify why and in case all other evaluations were made please consider including them (could be available in supplementary).
Discussion: (Related to the previous comment.) Also on that note, the evaluated concentrations are not reported in the discussion (only the toxic concentrations are mentioned with no direct relevance to IR).
Line 244: Please include the relevant reference
Author Response
This is good work done by the authors. Please consider the following suggestions:
Line 33: Kindly consider including the relevant reference here.
Response: We thank the reviewer for this comment. We have added the relevant reference in the revised manuscript.
Gregg, E.W.; Sattar, N.; Ali, M.K. The changing face of diabetes complications. The Lancet Diabetes & Endocrinology 2016, 4, 537-547, doi:https://doi.org/10.1016/S2213-8587(16)30010-9
Line 45: Kindly consider rephrasing “the drug” as no particular drug has been reported in the previous sentences. Also here the authors may want to further elaborate on why the choice of hepatocellular carcinoma cells is also appropriate for the evaluation of IR based on previous research.
Response: We thank the reviewer for pointing this out. We have revised as “The HepG2 cell line has the functional and physiological characteristics of normal hepatocytes and thus it has been used as an important model for exploring insulin resistance in vitro [10].”
Line 46-47: Please rephrase this sentence to make sure it is grammatically correct and conveys the authors’ intended message.
Response: We thank the reviewer for pointing this out. We have revised as “The HepG2 cell line has the functional and physiological characteristics of normal hepatocytes and thus it has been used as an important model for exploring insulin resistance [10]. According to previous studies, glucocorticoids have been noted to exacerbate hyperglycaemia in T2DM patients by destroying glucose homeostasis[11].”
Line 64-66: Please rephrase this sentence to make sure it is grammatically correct and conveys the authors’ intended message.
Response: Thanks for the comment. We have revised the sentence to “3,3’,4,5’-tetramethoxy-trans-stilbene (3,3’,4,5’-TMS) is a methoxy derivative of resveratrol. 3,3’,4,5’-TMS shows superior pharmacokinetic characteristics to remedy the shortcomings of resveratrol including low bioavailability and short half-life”.
Line 96: Kindly consider including the concentrations evaluated in this study. It would be beneficial for the reader to have this information right at the beginning in order to properly follow the development of the manuscript.
Response: Thanks for the suggestion. We have added the concentrations (1 μM and 2.5 μM) in the revised manuscript.
Line 100-111: Kindly include the original reference of the protocol employed by the authors (in case this is a protocol provided by the MTT supplier please consider including this information as well). It is essential for the reader to be able to understand that this is not a newly developed protocol for the evaluation of cell viability.
Response: We thank the reviewer for pointing this out. This is a protocol provided by the MTT supplier and we have added this information in the revised manuscript.
Line 104: Kindly elaborate on the selection on the 48h treatment as well as the number of cells seeded in each well (Considering that the volume of each well is approx.. 300μl and there is no report that the cell medium is refreshed within this 48h treatment, did the authors observe cell death ought to the number of cells increasing in each well?). Maybe a brief elaboration of the choices made on this protocol would be beneficial for the reader as well as future research.
Response: We gratefully appreciate for your valuable suggestion. As suggested by the reviewer, we elaborated the choices of incubation time and number of cells seeded. “The time of incubation for 48 h in MTT assay was selected as HepG2 cells were exposed to DXMS plus D-glucose to induce insulin resistance in latter experiments. The number of cells seeded was selected after screening the different cell densities according to the linearity range, at which no cell death was observed.”
Line 111: Kindly consider including the calculations made for the evaluation of cell viability.
Response: We thank for the valuable suggestion. “The absorbance was read at 570 nm using SpectraMax M5 microplate reader (Molecular Devices, Silicon Valley, CA, United States).” was described in section 2.3. To be more clear and in accordance with the reviewer concerns, we have added “The relative cell viability was calculated as the percentage of control group.”
Line 115 and 126: Please consider including the product number of the kit used.
Response: We thank the reviewer for this comment. As suggested by the reviewer, we have supplemented the product number of the glucose assay kit, A154-1-1.
Line 119 and 130: kindly consider describing the process in brief for the reader to understand what was done.
Response: We thank the reviewer for pointing this out. As suggested by the reviewer, we have expanded the description of the methods.
“The contents of cell glucose were determined by glucose oxidase method which causes a amaranth tint in the sample.”
“According to the manufacturer’s instructions, the cells were collected in phosphate buffer saline solutions with cell disruption by ultrasonication. After further hydrolysis in alkali solution, the contents of cell glycogen were determined by sulfuric acid-anthrone method which causes a blue tint in the sample.”
Line 130: Also include any calculations made for the final evaluation as well as the basis of the calculations (are they made against the control?)
Response: We gratefully appreciate for your valuable suggestion. They were made against the control which was indicated in the revised manuscript. “The cell glycogen levels were calculated by comparing with the control group.”
Line 135: Also here please include the specific treatments employed.
Response: We gratefully appreciate for your valuable suggestion. Treatments of the cells were same as above in section 2.2. “The cells were cultured with Low-glucose Dulbecco’s modified Eagle’s medium (DMEM) supplemented with 10% fetal bovine serum (FBS), penicillin-streptomycin, and non-essential amino acids (NEAA) which were obtained from Gibco (Carlsbad, CA, USA). The cells were maintained at 37℃ in a 5% CO2 humidified incubator. In order to establish IR cell model, the HepG2 cells were pretreated with DXMS (30 µM, dissolved in ethanol) plus D-glucose (50 mM final concentration) for 48 h. Thereafter, the experimental groups were treated with different concentrations (1 and 2.5 μM) of 3,3′,4,5′-TMS for 16 h. The group not treated with DXMS, D-glucose or 3,3′,4,5′-TMS served as the control.”
Line 133: The authors may want to consider including relevant references to the protocols employed if they are not provided by the supplier of kits used. In case this method is reported for the first time and no previous reference is available, please include all steps that led to the method’s development and standardization.
Response: We thank for this suggestion. This method is not reported for the first time and it is provided by the supplier of glycogen assay kit. We have added product number of the kit and the process was briefly described in the revised manuscript.
Line 153: Please include the positive control used for this evaluation. Also, the comment regarding line 133 applies here, so the authors are kindly invited to provide the reference of the original protocol or the steps that led to the method’s development and standardization if this is a newly introduced method.
Response: We thank the reviewer for this comment. We agree that the positive control could make the research better. However, we did not test and compare with positive control. The main goal was to investigate whether 3,3',4,5'-TMS has inhibitory effect on ROS level in IR-HepG2 cells.This is not a newly introduced method. CM-H2DCFDA is a general oxidative stress indicator. Determination of ROS level by CM-H2DCFDA is well established and widely used analytical method. As suggested by the reviewer, we included the reference in the revised manuscript.
Oparka, M.; Walczak, J.; Malinska, D.; van Oppen, L.M.P.E.; Szczepanowska, J.; Koopman, W.J.H.; Wieckowski, M.R. Quantifying ROS levels using CM-H2DCFDA and HyPer. Methods 2016, 109, 3-11, doi:https://doi.org/10.1016/j.ymeth.2016.06.008.
Line 176: Since both ethanol and DMSO are known to be toxic in cells in certain concentrations. Please include the volume of DMSO and ethanol used in this assay.
Response: We gratefully appreciate for your valuable suggestion. We indicated “0.25% v/v DMSO or 0.3% v/v ethanol” in the revised manuscript.
Line 182, 195 and 204: Please include the mean (SE) values in the text as the graphs are not very helpful for the reader to obtain this information.
Response: We gratefully appreciate for your valuable suggestion. As suggested by the reviewer, we have added the mean values in the text.
Line 216: Figure 4 (a) In the image for IRS1 the lines seem to be bending. Kindly revisit.
Response: We gratefully appreciate for your valuable suggestion. To be more clear and in accordance with the reviewer concerns, we have changed the representative image of IRS1 in the revised manuscript.
Results: Kindly elaborate on the limitation of the concentrations evaluated after the MTT assay. As the authors stated 0.5-10μM were not toxic, but further on only 1 and 2.5 are evaluated. Please clarify why and in case all other evaluations were made please consider including them (could be available in supplementary).
Response: We thank the reviewer for this comment. As shown in results on the glucose consumption and glucogen synthesis, we found that 2.5 µM is effective in ameliorate glucose consumption and glycogen content comparable to the control. Thus testing on higher concentrations at 5 and 10 µM may not be necessary.
Discussion: (Related to the previous comment.) Also on that note, the evaluated concentrations are not reported in the discussion (only the toxic concentrations are mentioned with no direct relevance to IR).
Response: We thank the reviewer for this comment. To be more clear and in accordance with the reviewer concerns, we have added the evaluated concentrations in the discusstion.
Line 244: Please include the relevant reference
Response: We thank the reviewer for this comment. As suggested by the reviewer, we have added the relevant reference.
Bugianesi, E.; McCullough, A.J.; Marchesini, G. Insulin resistance: a metabolic pathway to chronic liver disease. Hepatology 2005, 42, 987-1000.
Round 2
Reviewer 1 Report
accept as it is